# Barriers to and Facilitators of Technology Adoption in Emergency Departments: A Comprehensive Review

**DOI:** 10.3390/ijerph22040479

**Published:** 2025-03-23

**Authors:** Ann Thong Lee, R Kanesaraj Ramasamy, Anusuyah Subbarao

**Affiliations:** 1Faculty of Management, Multimedia University, Cyberjaya 63100, Malaysia; lee.ann.thong@student.mmu.edu.my (A.T.L.); anusuyah.subbarao@mmu.edu.my (A.S.); 2Faculty Computing Informatics, Multimedia University, Cyberjaya 63100, Malaysia

**Keywords:** technology adoption, influence factors, medical technology, barriers, healthcare, emergency, emergency room, ER, emergency departments, ED, acute care, emergency care, emergency medicine, accident and emergency, A&E

## Abstract

Background: Even while technology is advancing quickly in many areas, the healthcare industry, particularly emergency departments, is slow to incorporate new technologies. The majority of research is on healthcare in general, with few studies examining medical officers’ adoption of technology in emergency departments. Methods: This study used a comprehensive review design and examined a total of 30 peer-reviewed articles that were published between 2019 and 2024. The articles were reviewed by using keywords such as “technology adoption”, “influence factors”, “medical technology”, “barriers”, “healthcare”, “emergency departments”, “ED”, and so on. This review aimed to identify barriers and facilitators to provide insights to improve technology adoption in emergency departments. Results: The studies were conducted using different techniques, including surveys, interviews, and systematic reviews, to examine technology adoption in emergency departments across different geographic locations. The technologies studied include clinical decision support systems, telemedicine, electronic health records, and AI-based innovations. Several barriers were discovered in this study, including high employee turnover, accessibility issues, insufficient technology availability, resistance to change, and excessive workload. Key enabling facilitators were also identified, namely, good collaboration and communication, a supportive and engaged management team, and rigorous education and training. Conclusions: This study highlights that tailored strategies and collaboration are essential to overcoming barriers in emergency departments, which will lead to faster adoption of technologies that improve patient outcomes and efficiency. Further research will involve performing a deeper study of these findings and investigating more creative techniques to improve technology integration and further establish higher standards of care inside emergency departments.

## 1. Introduction

In recent years, due to the rapid development of technology, many industries such as the food and beverage and transportation industries, which are vital to human civilisation and have an impact on many aspects of daily life, have made significant progress and become more convenient [1,2,3]. However, the healthcare industry, another industry that is vital to humanity, although closely related to human health and life safety, is relatively lagging behind in the adoption and application of science and technology [4].

The healthcare industry has an extremely heavy workload due to its uniqueness. Healthcare professionals such as medical officers frequently deal with excessive workloads in the absence of technology, which is a widespread issue around the globe [5,6]. This is because of the reluctance of the healthcare industry to adopt new technology, which has caused many advanced medical innovations to move slowly in real-world applications [7]. Electronic health record (EHR), clinical decision support system (CDSS), and AI-driven diagnostic tools are some examples of these cutting-edge medical innovation technologies. These technologies have the potential to increase patient safety, decrease medical mistakes, and streamline workflows. They could be particularly useful for facilitating quick decision-making and cooperation in hectic emergency departments. For instance, CDSSs can help medical personnel make choices quickly, and EHRs make patient data conveniently accessible, guaranteeing that better and more thorough solutions can be suggested. However, although many cutting-edge technologies have been demonstrated to dramatically increase the effectiveness and quality of medical services, for a variety of reasons, their adoption has not followed the same trajectory as other industries [7].

As one of the most important departments in a hospital, the emergency department has been found to be relatively resistant to the introduction of new technologies in previous studies. This is mostly because of the unique challenges it encounters. First of all, medical officers working in the emergency department are too busy to devote time to application-oriented training [8]. As a result, little time is allotted for system adaptation, which results in a drawn-out adoption process because of the disparities in emergency medical officers’ technological proficiency [9,10]. Additionally, due to the fact that the emergency department has to treat a high number of patients [11], the implementation of new technology may disrupt current workflow and cause delays in treatment [12], particularly in emergency scenarios when unfamiliar technologies may cause a delay in making crucial judgements. Furthermore, the intricacy of situations presented in emergency departments necessitates adaptable technology, which frequently prompts questions about how usable cutting-edge technology will be [13,14]. Lastly, medical officers in the emergency department may exhibit resistance to change because they may be hesitant to give up tried-and-true techniques that have worked well in high-risk circumstances [15,16].

According to Chen et al. (2020) [17], medical officers exhibit poor adoption of new technologies, even though academics typically agree that technology has major benefits for the healthcare industry. However, the perspectives of medical officers have a significant influence on how technology is used in the medical industry as they are the ones who are directly involved with these innovations [18].

Although several studies have sought to evaluate the variables that encourage medical officers to adopt new technology, the majority of them have examined the healthcare sector as a whole rather than concentrating on specific parts of the industry. However, different perspectives on the need for new technological applications have resulted from differences in sectoral priorities. Therefore, the purpose of this study is to perform a thorough analysis of the variables that facilitate and hinder the adoption of technology in emergency departments and to determine how these factors influence the growth of technology in the healthcare industry.

By identifying and eliminating the barriers to technology adoption in the emergency department and exploring variables that may enhance technology acceptance, this study aims to increase medical officers’ acceptance of new technologies, as well as to accelerate technological advancement in the healthcare industry, especially in emergency departments. In addition, it is anticipated that the findings of this study will be a valuable resource for further research in this area, as this detailed approach could more clearly differentiate the challenges faced in different emergency department settings and provide insights specific to these unique settings. Furthermore, the study’s findings will also provide medical policymakers with a solid scientific foundation for developing more effective technology adoption strategies, as well as valuable feedback to technology developers on the usability and design of their products.

For this study, two research questions have been established:

What are the key barriers and facilitators influencing the adoption of technology in emergency departments?What strategies may assist emergency departments in successfully integrating health technologies to increase productivity and improve patient outcomes?

## 2. Literature Review

As technology advances, workflows are continually adapting in order to incorporate these innovations. However, adopting new technology may be extremely difficult in some industries, the most notable being healthcare. This is frequently because of the special requirements and the environment that this industry has.

The healthcare industry is among those with the most challenges when it comes to technology adoption. Despite the industry’s vital role in public health, healthcare workers have difficulty adjusting to new technology [19]. The adoption and use of technological advancements in healthcare organisations have been impacted by a number of variables that have come together to limit the introduction of technology. Numerous studies have examined the barriers to medical technology adoption by employing various approaches and examining the barriers and facilitators in technology use from different perspectives. However, the advantages of using technology are not clear, even though these studies offer deep insights and encompass the whole healthcare system. This might be because various departments deal with different kinds of challenges.

Emergency departments, being one of the departments that regularly experience work overload in the healthcare industry, deal with particularly complicated technology application challenges due to their high-pressure emergency work environment [20]. For example, given the enormous volume of work in emergency rooms, many technologies that function effectively in standard medical settings could be challenging to deploy successfully. Therefore, a more thorough knowledge of the obstacles and enablers of technology adoption may be obtained by carefully integrating studies relating to emergency departments.

As a result, this study will concentrate on the particular difficulties and potential opportunities of implementing technology in emergency departments. It will also examine the main obstacles to technology acceptance in this setting and evaluate efficient methods for encouraging technology use. This study seeks to give a more in-depth analysis to support the further development of technology applications by thorough assessment and summary of the literature related to emergency departments in the healthcare industry.

### Reviewed Studies

Hall et al. (2022) [21] conducted a study that evaluated the introduction of virtual urgent care (VUC) services in Ontario, Canada, and identified several facilitators and barriers. According to their study, the facilitators included patient engagement through the integration of feedback to address needs, provincial funding that supported the launch and sustainability of programmes, local champions who guided the effective delivery of programmes, and successful marketing strategies that promoted awareness and encouraged participation. Meanwhile, they also identified barriers, which were the need for behaviour change strategies to assist healthcare providers and patients in adjusting to virtual care, challenges with IT infrastructure and system integration, ensuring equitable access to services for all patients regardless of socioeconomic status, and the need for standardised data collection to assess programme impact and effectiveness. The study indicated that even though the VUC programmes were designed to expeditiously establish safe medical care and divert low-acuity patients from emergency departments during the COVID-19 pandemic, regional autonomy led to customised solutions that presented challenges for long-term sustainability and future funding. Finally, the study’s findings highlight the importance of ongoing quality control and data collection in ensuring the long-term survival of programmes.

Fujimori et al. (2022) [22] used a mixed methods approach to assess the acceptability, barriers, and facilitators of these systems, integrating quantitative and qualitative assessments. In this study, the respondents were 14 doctors from two community tertiary care hospitals in Japan using a real-time AI-based CDSS intended to predict aortic dissection. By conducting this study, several important barriers were found, which included worries about alert fatigue and system performance, compatibility problems like typing speed impacts, and system failure anxiety. On the other hand, the study also revealed key facilitators, such as the quality of design, with an easy-to-use interface and real-time alerts boosting usability, and the strength of evidence, as trust in the system’s data properly mirrored local illness patterns. By conducting this study, they highlighted that enhancing system performance and system compatibility while using an evidence-based design and user-friendly interfaces is essential for gaining excellent user acceptability of AI-based CDSSs in emergency departments.

Anaraki et al. (2024) [14] conducted a study to examine the barriers and facilitators associated with implementing SurgeCon, a quality improvement programme designed to boost efficiency and patient satisfaction in emergency departments (EDs) throughout Canada. In this study, they included multiple hospitals from both urban and rural areas in order to capture a range of experiences. By conducting this study, they revealed several important facilitators, such as engaged and supportive management, sufficient personnel and resources, efficient departmental communication, successful prior intervention experiences, and a strong drive for progress. Effective supervisors foster success by encouraging employees and guaranteeing resources, and positive reinforcement of new endeavours stems from effective communication and past achievements. On the other hand, they also discovered barriers including limited management participation, authoritarian leadership, a lack of personnel, frequent turnover, inadequate communication, a lack of cooperation, unfavourable prior experiences, aversion to change, and excessive workloads that impede advancement and incite opposition among employees. This research shows that a comprehensive understanding of the facilitators and barriers associated with healthcare quality improvement programmes is crucial for their successful implementation. Besides that, the study also suggests strategies including employee education, appointing advocates to spearhead projects, and encouraging proactive participation in order to overcome barriers and enhance emergency department effectiveness and patient outcomes.

Huilgol et al. (2024) [23] examined the difficulties faced and the ways in which emergency department physicians made decisions in response to novel developments during the COVID-19 pandemic. A total of 49 healthcare professionals, including 17 doctors, 7 advanced practice providers, 18 nurses, and 7 respiratory therapists, participated in the study from eight hospital-based emergency departments in the United States. By conducting this study, they identified several facilitators of innovation, including the use of social media, clinician autonomy, organisational culture, supportive leadership, and external experiences. On the other hand, barriers included a dearth of information supported by research, evolving recommendations, anxiety, moral discomfort, and clinical opposition. According to the study’s findings, organisations may foster innovation by designating capable leaders, guaranteeing the psychological stability of their staff, offering the required procedures and resources, and acknowledging accomplishments.

Zachrison et al. (2020) [24] conducted a study aimed at understanding the barriers to telemedicine implementation in rural emergency departments (EDs) and analysing the characteristics of rural EDs that do and do not use telemedicine. Data from the 2016 National Emergency Department Inventory (NEDI) survey as well as follow-up questionnaires were used in the study. Through this study, they discovered several barriers, the most important of which was cost, which was mentioned by 37% of rural EDs. Besides that, staffing concerns, transfer patterns, and the perception of telemedicine system complexity were additional barriers. Meanwhile, this study also found the key facilitators of telemedicine adoption, which were support from the hospital, EDs, and health system leadership. The results of this study reveal the need to address financial restrictions as well as other challenges in order to increase telemedicine adoption and improve rural healthcare access.

Boyle et al. (2023) [25] conducted an investigation that focused on the barriers, facilitators, and readiness of New England hospitals in the United States to implement a regional catastrophe teleconsultation system intended to provide quick access to medical specialists in times of crisis. In this study, a cross-sectional survey was conducted either via phone or online with emergency managers from hospital-based and freestanding emergency departments (EDs) across New England. One of the main barriers that was noted was the difficulty of reaching burn, toxicological, radiation, and trauma specialists, which could cause delays in receiving critical treatment. Another significant barrier was the potential for credentialing delays, which might impede expert consultations. A total of 70% of hospitals required catastrophe credentialing, which could take up to 72 h. Additionally, the usage of the technology was further complicated by the unreliability of cell-phone connectivity and internet in rural hospitals. Fortunately, the majority of the hospitals supported the deployment of the system by having sufficient emergency alerting systems and telecommunications equipment. Furthermore, considering the advantages of quick expert access, 81% of hospitals, especially smaller and rural ones, expressed a strong readiness to employ teleconsultation technology. The results of the study showed that, despite the system’s great demand, improvements in communications redundancy and consistent credentialing procedures are crucial to both its successful deployment and capacity to optimise benefits in emergency scenarios.

Pu et al. (2024) [26] examined the increasing usage of virtual emergency departments (VEDs) in Victoria, Australia, particularly during the COVID-19 pandemic, and found a number of barriers and facilitators for their adoption. The study investigated the benefits, shortcomings, and scalability of VEDs through interviews with 20 individuals, including emergency medicine doctors, healthcare providers, and other experts. By conducting this study, several key barriers were the challenge of performing comprehensive virtual physical exams, the deficiency of solid data and standardised protocols for VEDs, and the increased workload and resource limitations experienced by emergency department personnel. On the other hand, they also identified several facilitators, such as the ease of use and availability of VEDs for patients and medical professionals, better follow-up and coordination, and increased safety through lowered infection risks and lessened patient anxiety. In conclusion, the findings of this study revealed that although the virtual evaluations (VEDs) can be a useful and practical substitute for in-person visits, they are not a perfect substitute because they still have limitations in real-life use, particularly for the user, and especially older adults receiving residential care. Therefore, further research, the development of standards, and the provision of resources are needed to enable the wider application of VEDs.

Antor et al. (2024) [27] performed a study to assess the adoption and usability of Electronic Health Records (EHRs) at Ghana’s Komfo Anokye Teaching Hospital, focusing on variables influencing adoption and obstacles experienced by healthcare workers. Using a cross-sectional survey, healthcare personnel shared their experiences with EHRs. This study found several barriers that impede efficient healthcare delivery, such as inadequate system training, frequent malfunctions, power outages, privacy issues, and poor maintenance. Positively, they also discovered that the facilitators that encouraged EHR adoption were system comfort, dependability, and higher-quality patient care. To sum up, this study highlights the importance of EHRs in improving healthcare and also emphasises how barriers must be overcome and how EHR advantages must be fully realised through continual training, technical assistance, and infrastructure updates.

An investigation of the technology barriers and facilitators in providing paediatric mental and behavioural healthcare in emergency departments (EDs) was carried out by Bhosekar et al. (2023) [28]. Through semi-structured interviews and observational studies involving four healthcare professionals from two different institutions, this study identified several significant barriers and facilitators. According to their study, barriers included worries over the calibre of third-party services like telehealth platforms, poor training on new software capabilities that led to underutilisation, and software usability difficulties such as complicated interfaces and frequent changes that doctors found difficult to traverse. On the other hand, effective safety measures that guaranteed the protection of patients and staff, easily accessible clinician notes that enhanced continuity of care, and effective communication systems that simplified information exchange among healthcare practitioners were the facilitators that were found in this study. According to this study, although technology can improve treatment, there are drawbacks that need to be taken into consideration when employing technology in order to increase patient safety and reduce physician burnout. In conclusion, the results of this study suggest that regular training and human-centred design might reduce these barriers and enhance the process of providing treatment in emergency departments.

In order to improve access to treatment for marginalised patients in Toronto’s emergency departments (EDs) and reduce digital health inequities, Hodwitz et al. (2024) [29] conducted a study to evaluate a hospital-based phone prescription scheme. A total of 12 healthcare professionals were interviewed for the study, which focused on five main goals, which were establishing patient trust, giving patients the authority to manage their own care, bridging gaps in the system, providing equitable treatment for marginalised communities, and reducing moral distress among personnel. However, the programme faced challenges such as unclear eligibility criteria, inconsistent phone availability, and potential bias in administration. On the positive side, healthcare staff’s non-judgemental, anti-oppressive approach, a smooth enrolment process, and deep understanding of marginalised patients’ needs were important facilitators. The study concluded that providing phones to marginalised patients can help bridge gaps in both digital and social health, but its success depends on building trust, understanding patients’ unique needs, and adopting a holistic, biopsychosocial approach to healthcare.

The factors influencing the uptake and sustainability of telehealth systems in rural emergency departments in the United States were examined by Nataliansyah et al. (2022) [30]. They identified numerous barriers and facilitators by conducting semi-structured interviews with 18 important informants from six different healthcare systems. One of the barriers was the lack of adequate needs assessments, which led to services that did not meet the unique requirements of rural areas. In addition, the lack of proper training for local employees resulted in problems with unsuccessful consultations since they were not familiar with the technology. Furthermore, the other barrier was that the regular provision of telehealth services in remote hospitals was impeded by resource limits. Aside from these impediments, this study identified other barriers caused by political and regulatory difficulties. Additionally, they discovered that the facilitators included in-depth needs evaluations, guaranteeing that the services were customised to satisfy regional needs. Besides that, thorough staff training initiatives contributed to the efficient application of telehealth technology, which was also considered one of the facilitators. Furthermore, strong ties between the hub and spoke locations fostered cooperation and trust, which are essential for operational success, and adequate service capacity and backup mechanisms allowed for more dependable service delivery. In conclusion, this study highlights the significance of addressing regional issues and focussing on these facilitators to ensure the effectiveness and sustainability of telehealth adoption.

Kennedy et al. (2024) [31] carried out a study to understand how the HIRAID^®^ emergency nursing framework was being used in rural emergency departments in Southern New South Wales, Australia. The research focused on exploring its application in these settings. Therefore, they used a mixed methods approach to poll 102 emergency nurses from 11 departments. The average experience of the participants was 16 years in nursing and 8 years in emergency care. By conducting this study, they discovered numerous barriers to the implementation of HIRAID^®^, such as time and resource restrictions, a lack of understanding of the framework, a lack of management support, and uncertainties over the framework’s efficacy. The nurses’ excitement for novel approaches and their belief that HIRAID^®^ may enhance nursing procedures and patient outcomes, on the other hand, were also recognised as facilitators. This study emphasised the importance of developing strategies to address the unique challenges faced in EDs, especially in smaller departments, to ensure the successful implementation of HIRAID^®^. It concluded that customised approaches are necessary to overcome resource limitations and provide essential support and training, while also tackling the specific difficulties of adopting new practises in rural emergency settings. The findings of this study were intended to lead to the development of a solid, realistic plan for implementing the HIRAID^®^ framework throughout the district.

The effects of workflow fragmentation and electronic health records (EHRs) on the amount of paperwork required in emergency departments (EDs) were investigated by Moy et al. (2023) [32]. Using the EHR from Epic Systems, the researchers conducted semi-structured interviews with 24 US emergency department physicians and registered nurses. The study revealed numerous barriers to efficient recordkeeping. Notable barriers include EHR shortcomings, such as inadequate functionality and a poorly designed user interface, which lead to an increase in manual labour and disruptions in workflow. Meanwhile, frequent job switching and interruptions exacerbate these difficulties by increasing cognitive strain and lowering the calibre of documentation. Additionally, the study found a number of facilitators who may assist in resolving these issues, which include incorporating patient-specific EHR features, minimising manual data entry through the use of sophisticated data collection techniques, and improving EHR displays and setups for improved usability. The study indicated that although EHRs improve patient care, in order to minimise paperwork load, their design has to be more closely linked with the unique requirements of emergency department processes. To determine if optimising the current EHR systems would be sufficient or whether a thorough redesign is necessary, further input from stakeholders is needed.

The rapid adoption of telehealth systems in American emergency departments (EDs) during the first nine months of the COVID-19 pandemic was studied by Uscher-Pines et al. (2021) [33]. They investigated the implementation of telehealth solutions using semi-structured interviews with 15 emergency department leaders from 14 institutions spread across 10 states. The leaders were selected via literature review and snowball sampling. The study revealed that prior experience with telehealth facilitated the swift implementation of virtual post-discharge evaluations, tele-isolation, tele-triage, and teleconsultation. However, many of these solutions were temporary as their relevance diminished over time. By conducting this study, they found a number of facilitators and barriers. Two key barriers were the need for direct support inside emergency departments and technological challenges. Facilitators included higher compensation, more lenient licensure standards, and loosened HIPAA rules. This study highlights how the pandemic sped up telehealth usage, though many projects were discontinued once demand decreased. The insights from this study could inform future disaster response planning and underline the importance of a robust telehealth infrastructure and ongoing support to address future logistical and technological challenges.

Wong et al. (2024) [34] developed a research procedure to assess emergency department-manage, a clinical decision support system (CDS) intended to manage agitation symptoms in emergency departments (EDs). This study aims to identify individuals who may become agitated and to assist emergency department physicians in treating these patients appropriately to reduce the need for restraints and enhance patient outcomes. The approach includes a formative assessment as well as qualitative data gathered from emergency department physicians, nurses, technicians, and patients who have been restrained. A pilot randomised controlled trial will be conducted at two adult emergency department locations in the Northeastern United States with the aim of enrolling a minimum of 26 eligible participants. Implementing optimal practises for agitation control might be difficult due to time restraints, inconsistent workloads, and restricted access to professional psychiatric evaluations. Meanwhile, a user-centred design approach that integrates end-user feedback continuously, proactive interventions, and the use of systematic risk assessments will likely be facilitators. The findings of this study suggest that if emergency department-TREAT is implemented successfully, it may be able to identify patients who are more likely to be at risk, reduce the need for restraints, and enhance patient outcomes. If successful, a follow-up clinical efficacy trial will compare emergency department-TREAT to standard care at other emergency department sites.

Barton et al. (2024) [35] conducted a study to explore how academic detailing could support the design and implementation of a clinical decision support (CDS) tool aimed at preventing falls in emergency departments. The study, which featured 16 semi-structured interviews with emergency medicine residents and advanced practice physicians who had used the CDS tool, was carried out at a sizable university medical centre in the Midwest of the United States. The study focused on individuals aged 65 and older. Several facilitators and impediments were identified that affected the use of the CDS tool. Barriers encompassed the hectic atmosphere of the emergency room, physicians’ misinterpretations of the CDS tool or the referral procedure, and the absence of clarity in the referral process. Facilitators, on the other hand, included the tool’s ease of use, minimal input requirements, and its automated identification of high-risk patients, enabling timely interventions. Academic detailing interviews were designed to clarify misconceptions and enhance understanding of the CDS tool and the referral process. The study’s findings show that academic detailing may successfully promote the adoption of health information technology by identifying the factors that affect utilisation and teaching physicians. This strategy allows for larger-scale redesigns and real-time changes, which helps to align technology to the evolving demands of the clinical workforce.

A qualitative interview study was conducted by Billah et al. (2022) [36] to investigate the use of patient decision aids (DAs) in emergency departments (EDs). A total of 20 emergency doctors from a variety of New York health systems, including attending physicians, residents, and physician assistants, participated in the study. The goal of the study was to determine what barriers and facilitators individuals presenting with low-risk chest discomfort and unexplained syncope face and benefit from while utilising DAs. The study identified six main barriers, which were patients’ worries, such as inadequate health literacy, uncertainties about the validity of the DAs, concerns about increased medicolegal risk, a perception that DAs are unnecessary, and lack of time to apply them. Meanwhile, the six positive attitudes regarding shared decision-making (SDM), patient access to follow-up care, the possibility of increased patient satisfaction, better risk communication, efficient integration of DAs into clinical workflows, and institutional support were found to be facilitators. Based on the identified facilitators and these barriers, the study indicated that enhancing the use of DAs in EDs is imperative. The findings of this study might influence future strategies to enhance the use of DAs and support standardised SDM in emergency care settings.

The difficulties and facilitators of introducing digital psychological therapies for senior citizens in emergency rooms (EDs) were examined by Davison et al. (2024) [37]. Their scoping review encompassed both qualitative and quantitative studies, adhering to the PRISMA-ScR checklist and the Joanna Briggs Institute recommendations. The research focussed on patient, family, and emergency department staff perspectives, attitudes, experiences, and perceptions regarding digital psychological therapies. The evaluation focussed on people 70 years of age and older, their families, and emergency department staff. Four databases, including Medline, Embase, PsycINFO, and Scopus, as well as the top 100 results from a Google Scholar search, were searched thoroughly. Several barriers were identified in the study, including the complexity of health problems older adults face, the need for care that is urgent, and the possibility that older adults are not tech-savvy. Meanwhile, facilitators were also identified, including the willingness of older individuals using portable electronic devices (PEDs) to evaluate themselves and the potential for these therapies to improve care and decrease unforeseen hospital visit. It is predicted that new psychosocial digital health technologies for use in emergency departments (EDs) will be created with the results of this scoping study as a guide to enhance the psychosocial evaluation and care of older adults.

Salwei et al. (2022) [38] investigated the usability problems and enhancers of a clinical decision support (CDS) system based on human factors engineering (HFE) to detect pulmonary embolism (PE) in emergency departments (EDs). In the study, 32 emergency care professionals, including attending doctors and residents, were debriefed via a scenario-based simulation. Through interviews and deductive content analysis using Scapin and Bastien’s usability criteria, the team identified 271 usability issues, comprising 94 barriers and 177 facilitators. Facilitators included features like the CDS’s ability to automatically display vital signs, saving doctors time in finding information, and its support in ordering diagnostic tests and generating paperwork. However, challenges arose from the system’s incompatibility with certain workflows, such as those of doctors preferring single risk assessments or simultaneous orders, as well as limited support for collaboration between residents and attending physicians. The study concluded that applying HFE principles improves usability in CDS design, but emphasised that better integration into existing workflows is crucial to overcoming usability challenges. The findings highlight the importance of considering both broad and detailed usability factors to ensure the successful adoption of CDS technology in clinical practice.

The parameters impacting the acceptance, use, and upkeep of a clinical decision support (CDS) tool for buprenorphine initiation in emergency departments (EDs) were investigated by Simpson et al. (2023) [39]. The study discovered several facilitators and barriers through 28 interviews with clinicians from five different healthcare systems who were participating in the EMBED experiment, including attending doctors, physician assistants, and residents. Barriers encompassed clinical training, organisational culture, patient referrals for further care, and customisation of implementation for each emergency department. In the meantime, several factors played a crucial role in facilitating the adoption of the CDS tool. These included the efforts of local activists, effective training programmes, a supportive environment, and efficient referral tools. This study revealed that while the CDS tool alone had a minimal impact, significant adoption benefits come from a comprehensive, multilevel approach. To truly enhance the effectiveness of CDSs in promoting evidence-based practises in emergency care, it is essential to tailor their implementation, ensure robust organisational support, and provide practical, hands-on training.

Shin et al. (2024) [40] examined the factors influencing the adoption of digital applications for suicide safety planning in a psychiatric emergency department. The researchers used the Theoretical Domains Framework (TDF) and the COM-B model (based on capability, opportunity, motivation, and behaviour) to conduct semi-structured interviews with a total of 29 emergency department professionals, including nurses, psychiatrists, social workers, programme assistants, and chemists. The app’s potential to enhance patient accessibility and care efficiency, as well as the physicians’ strong incentive to utilise it stemming from their feeling of professional identity and duty, were highlighted as key facilitators. However, adoption was hampered by problems with paperwork, communication, and patient access to cell phones. By conducting this study, they found that the lack of connectivity between the app and existing electronic health record systems posed a significant barrier to its easy integration into standard clinical processes. Besides that, this study also highlighted the importance of developing targeted strategies to overcome these barriers and capitalise on facilitating factors. Continuous evaluation and modification of these strategies will greatly facilitate suicide prevention programmes for psychiatric EDS patients, as new barriers may emerge and facilitating factors may change. This will guarantee the app’s long-term viability.

Shuldiner et al. (2023) [41] conducted a study in an Ontario hospital to investigate the creation and evaluation of a virtual emergency department (VED) prototype. The study evaluated VED acceptance, reliability of implementation, and impact on continuity, quality, and access to care. By conducting research, they found that providing a safe, quiet environment for virtual visits, quick and convenient access to treatment, and high patient satisfaction were all factors in VED success. The service’s iterative expansion approach is another key factor that enables it to adjust to the demands of patients and doctors. On the other hand, they discovered that barriers include problems in scheduling follow-up meetings and testing via the platform, as well as concerns among medical specialists that their experience would not be efficiently exploited in a distant setting. The VED has an average of 153 visits per month and has received an overall positive response, improving patient satisfaction and access to care. However, ensuring the long-term viability of the service depends on removing the relevant barriers and continuously improving the service.

The study undertaken by Sharifi Kia et al. (2023) [42] aimed to provide a comprehensive analysis of the benefits and challenges associated with the use of telemedicine in emergency departments (EDs). A total of 12 of the 18 studies that were included in the analysis had a high risk of bias after undergoing a thorough methodological quality evaluation. According to the analysis, nine studies support real-time video conferencing as the most recommended telemedicine technology for emergency departments. Meanwhile, eight of the studies cited cost savings as a major advantage, while six pointed out infrastructural and technological difficulties. The study found that increasing the general efficacy of emergency services requires telemedicine adoption for two main reasons, including improving patient care and reducing costs. Despite these advantages, there are still a lot of obstacles to be solved, such as poor infrastructure, difficult technology, and the need for substantial evidence to support viability. The study concluded that, despite telemedicine’s great potential to improve patient care, more extensive study is necessary before emergency rooms can employ it.

Hose et al. (2023) [43] explored the challenges and enablers of adopting health information technology (HIT) in team-based healthcare environments. They were able to identify key factors that influenced the deployment of HIT by conducting interviews with 36 healthcare workers in 12 different professions. The study revealed that major barriers include technical issues, such as system incompatibilities and software malfunctions, which can disrupt workflows and reduce efficiency, as well as inadequate training and resistance to change among staff. These barriers highlighted the necessity of solid technical assistance and ongoing training to guarantee that integration goes well. Positively, the study also found that strong leadership, sufficient money, and thorough training were essential for the successful implementation of HIT. It became clear that organisational commitment to technology and human resources was essential to bringing about this transformation. The study concluded that HIT can be made much more effective and efficient by utilising the discovered facilitators and eliminating the relevant barriers. Doing so would eventually improve patient care and streamline healthcare operations.

Tyler et al. (2024) [44] explored how artificial intelligence (AI) and machine learning (ML) can enhance triage processes in emergency departments (EDs). In the study, they examined a total of 1142 publications from databases such as EMBASE, Ovid MEDLINE, and Web of Science as part of a systematic review of the literature up to September 2023. After a comprehensive screening process, 29 publications were selected for in-depth analysis. The findings indicated that AI has significant potential for boosting the consistency, efficiency, and accuracy of patient assessments. Over time, these advancements could result in improved patient care and more efficient use of resources. Nevertheless, the study identified several barriers to the successful integration of AI in triage emergency departments. Among these barriers were staff resistance, worries about data security, high initial costs, and the need for algorithms that are clear and understandable. In conclusion, while AI offers exciting possibilities for emergency medical care, addressing these challenges is essential for fostering collaboration, building trust, and ensuring smooth integration into healthcare systems.

Hudson et al. (2023) [45] investigated the potential use of AI-driven social robots to reduce children’s anxiety and pain during intravenous procedures in paediatric emergency departments (EDs). A total of 11 medical professionals from two paediatric emergency departments in Canada were interviewed for the study. By conducting the study, they discovered several facilitators, which included procedure-related stress, which might be considerably decreased by providing each kid with individualised emotional support and distractions based on their preferences. Positive reinforcement from robots was also thought to be advantageous. Besides that, they also identified several barriers, which included the need for age-appropriate interactions, space constraints in EDs, and the capacity of the robots for independent situational adaptation. The study concluded that although lowering distress levels using AI-enhanced social robots might improve paediatric care, overcoming these barriers is necessary for widespread adoption. In order to achieve this, the involvement of healthcare professionals is critical to the development of effective robots that function effectively in busy emergency departments.

In a review by Katzman et al. (2023) [46], the expanding role of AI in emergency radiology was analysed through the lens of both interpretive and non-interpretive applications. They studied a total of 44 studies that addressed the use of AI in workflow optimisation, image quality assurance, picture protocol management, and common emergency condition diagnosis. The study focused on the possible benefits of AI, such as how it may speed up patient treatment and automate tedious tasks, which would reduce radiologists’ workload and improve outcomes. In the study, several significant barriers were identified to the widespread application of AI in radiology. These barriers included costly costs, the need for medical education, and doubts over the reliability of algorithms. The study suggested that both a financial investment and a cultural change in medical practice are necessary to overcome these challenges.

In order to examine the role of AI technologies in emergency medical treatment, Piliuk and Tomforde (2023) [47] conducted a systematic evaluation of 116 studies from 380 publications gathered from reputable databases such as IEEE Xplore, ACM Digital Library, Springer Library, ScienceDirect, and Nature. The focus of the selected articles was on using machine learning and deep learning techniques in emergency medical services. By conducting the study, they discovered that AI can handle large volumes of data quickly, which improves diagnosis, optimises patient flow, enhances decision-making, and manages resources more effectively. These benefits might lead to better patient outcomes. However, the study also revealed several barriers, including the expensive cost of AI, the need for extensive training, concerns about data privacy, and unequal AI performance in different clinical settings. The study concluded that further quality, prospective research is required to validate the benefits of artificial intelligence and address the challenges associated with integrating it into emergency medicine.

Jordan et al. (2023) [48] studied the introduction of KATE, an artificial intelligence clinical decision support tool, into the emergency department triage system of a community hospital in the United States. A total of 13 triage nurses from the emergency room participated in the study, and semi-structured interviews were used to gather data. Initially, the AI tool was considered with distrust due to its purported inability to take into account the contextual requirements of the scenario. However, as time passed, the nurses began to regard the tool as a helpful aid for supporting decisions. The study revealed several critical facilitators for its successful adoption, including enhanced patient safety, efficient assistance for decision-making, thorough training, and easy integration into current processes. However, barriers included aversion to change, a reliance on clinical expertise, and scepticism over the AI system’s capacity to take into account contextual and cultural elements. The study emphasised the significance of taking clinical staff knowledge and cultural aspects into account while deploying Al solutions in healthcare settings.

In order to help the reduce congestion in physical emergency departments, Australia established the Victorian Virtual Emergency Department (VVED) to provide virtual care for non-emergency situations. Talevski et al. (2024) [49] examined the programme’s development and success. Over 300,000 patients in Victoria have received consultations via the VVED since its introduction, with an average of over 600 virtual consultations every day. The study findings indicated that several factors contributed to the programme’s efficacy, such as robust institutional backing, progress in digital healthcare, and the need for substitute care approaches amid the COVID-19 outbreak. However, barriers included technical difficulties, resistance from healthcare professionals, and concerns about the quality of care provided virtually. Although virtual emergency departments (VEDs) can significantly reduce overcrowding and improve patient outcomes, the study, which used 500 patients who used the VVED service as a sample, concluded that continued improvements are necessary to ensure the feasibility and usefulness of such programmes.

## 3. Materials and Methods

### 3.1. Research Methodology

Throughout the thorough assessment, several criteria were used to guarantee the quality of this study. Firstly, the main goal of this study was to include existing research on the factors that encourage and hinder the adoption of new technologies, particularly in emergency departments. As such, 30 peer-reviewed articles relevant to the use of technology in emergency departments were included. There was a significant variation in the sample sizes of these articles since they were included regardless of their sample size. This is due to the fact that qualitative research with small sample sizes may effectively provide in-depth insights into specific challenges, whereas quantitative studies with large sample sizes may provide more broad statistical trends. Therefore, during the selection process, this study gave more weight to methodological soundness, depth, and results’ applicability to emergency department technology usage than just sample size. This approach ensured a comprehensive understanding of the use of technology in emergency departments.

Secondly, a comprehensive search was carried out with keywords like “technology adoption”, “influence factors”, “medical technology”, “barriers”, “healthcare”, “emergency”, “emergency room”, “ER”, “emergency departments”, “ED”, “acute care”, “emergency care”, “emergency medicine”, “accident and emergency”, and “A&E” to make sure that only relevant articles were included.

Third, only articles that were published between 2019 and 2024 were taken into account in this study to ensure the review incorporates the most recent research and the latest data on the subject at hand.

Lastly, the reviewed articles were gathered from credible and applicable sources, such as the JMIR, Frontiers, ScienceDirect, BMC Health Services Research, SpringerLink, ScienceDirect, MDPI, BMJ, Research Square, BMJ, NLM, and Sage Journals.

The PRISMA flow diagram for this study is shown in Figure 1, which outlines the procedure followed in locating and choosing pertinent papers for inclusion.

In order to ensure that only relevant studies were included, this study began by conducting a database search using keywords such as “technology adoption”, “influence factors”, “medical technology”, “barriers”, “healthcare”, “emergency”, “emergency room”, “ER,” “emergency departments,” “ED,” “acute care,” “emergency care,” “emergency medicine,” “accident and emergency,” and “A&E”. A total of 108 records were identified through database searches from sources such as Frontiers (*n* = 7), JMIR (*n* = 7), BMC (*n* = 6), ScienceDirect (*n* = 13), SpringerLink (*n* = 9), JAMIA (*n* = 9), BMJ (*n* = 12), Research Square (*n* = 11), MDPI (*n* = 7), NLM (*n* = 9), Sage Journals (*n* = 7), JACEP OPEN (*n* = 5), Nature (*n* = 3), and IEEE (*n* = 3). From these 108 records, studies that did not meet the criteria were eliminated, such as duplicate studies, abstracts that did not relate to the emergency department or the adoption of technology, and studies published outside of 2019 to 2024.

First, 8 studies were eliminated due to duplication. Thus, only 100 records in total remained to be reviewed. A total of 40 studies were then eliminated because their abstracts or titles did not discuss the use of technology in emergency departments. Following that, 40 studies had to be excluded after assessing the eligibility of the remaining 60 studies. From these 40 studies, 18 studies were eliminated for concentrating on other departments, 15 studies were eliminated for not being related to technology, and 7 studies were eliminated due to their published time frame, as they are published outside of 2019 to 2024. Subsequently, an additional 10 studies were incorporated into the review process. This resulted in an increase in the total number of studies selected for the final analysis up to 30. This was done since bigger sample sizes are able to yield more reliable results. Due to this systematic process, only the most relevant publications were kept for examination.

### 3.2. Meta-Analysis

#### Narrative Synthesis (Qualitative Meta-Analysis)

This study was unable to employ conventional meta-analysis techniques due to the lack of specific quantitative results as well as effect estimates, statistics, and raw data in the reviewed articles. To explore the barriers and facilitators that affect technology adoption in emergency departments, this study employed a narrative synthesis technique.

The evaluated studies contained both qualitative and quantitative methodologies, and there was a wide range in sample sizes and methodology. For example, a study conducted by Hall et al. (2022) [21] examined the deployment of virtual emergency care with 13 participants. On the other hand, a study carried out by Fujimori et al. (2022) [22] assessed the efficacy of an AI-based CDSS using a sample size of 27,550 patients. The range of sample sizes demonstrates the diversity of study environments, ranging from small-scale qualitative investigations to extensive quantitative assessments.

The results of the studies that were analysed in this study reveal the common barriers and facilitators to the technology adoption in the emergency department, notwithstanding the absence of precise figures. Numerous studies have revealed the presence of barriers such as high turnover rates, limited personnel training, and system incompatibility. For instance, Zachrison et al. (2020) [24] emphasised the expense and complexity of telehealth systems as important barriers facing rural emergency departments. Meanwhile, Anaraki et al. (2024) [14] underlined low management participation as a major barrier. Further, Fujimori et al. (2022) [22] discovered that system incompatibility is one of the key barriers that influence novel technology acceptance. On the other hand, management that was supportive, good at communicating, and focused on leading with strength was an often-mentioned factor that helped EDs embrace technology. Studies such as Huilgol et al. (2024) [23] and Kennedy et al. (2024) [31] supported this perspective. Their studies demonstrated that the successful implementation of new technologies in emergency departments requires strong leadership and an adequate workforce.

Although quantitative meta-analysis was unable to be employed for this study, the narrative synthesis provides a comprehensive understanding of the ongoing barriers and facilitators that influence technology use in emergency departments. The results suggest that strengthening leadership support, staff training, and system usability is crucial to overcoming hurdles and expediting the adoption of technology developments in emergency departments.

## 4. Discussion

### 4.1. Literature Analysis

Table 1 presents an overview of 30 studies examining the adoption of technology in emergency departments (EDs) across various global healthcare settings, including Canada, the United States, Australia, Japan, and Ghana. These studies cover a diverse range of technologies, such as telemedicine, virtual emergency departments (VEDs), clinical decision support systems (CDSSs), artificial intelligence (AI)-driven solutions, and electronic health records (EHRs). The research methodologies employed include qualitative approaches (interviews, case studies), quantitative methods (surveys, statistical modelling), and mixed-method research, ensuring a comprehensive understanding of technology adoption challenges. The sample sizes vary significantly, with some studies analysing small groups of 10–20 participants, while others assessed thousands of cases. The study participants included emergency physicians, nurses, hospital administrators, IT specialists, and patients, providing a multi-stakeholder perspective on the barriers and facilitators influencing technology adoption.

A key trend observed in these studies is the difference in challenges between rural and urban emergency departments. Studies focusing on rural EDs highlight limited infrastructure, high implementation costs, and staffing shortages as primary barriers to technology adoption, particularly for telemedicine and virtual care solutions. In contrast, urban EDs face challenges related to system integration, staff resistance to AI-based decision support tools, and concerns about data security and privacy. Common barriers across all settings include high workload, lack of training, and resistance to change, whereas facilitators include supportive management, strong leadership, user-friendly system design, and well-structured training programmes. The findings suggest that tailored strategies, including increased financial investment and enhanced training, could improve technology adoption in emergency departments and ultimately enhance patient care and operational efficiency.

Table 2 provides a detailed comparison of the barriers and facilitators influencing the adoption of technology in emergency departments (EDs) across 30 reviewed studies. The barriers identified in these studies highlight the challenges that hinder technology adoption, including high implementation costs, insufficient staff training, resistance to change, workflow disruptions, lack of IT infrastructure, and concerns about data security. In rural emergency departments, resource constraints, poor internet connectivity, and limited access to specialists are major barriers, particularly for telemedicine and virtual emergency care solutions. In urban settings, barriers are often related to usability issues, system compatibility, and physician scepticism toward AI-based decision support tools (such as CDSSs). Additionally, high staff turnover, excessive workloads, and lack of institutional support further delay technology integration in many emergency settings.

Besides that, Table 2 also highlights key facilitators that enhance technology adoption in emergency departments. Strong leadership support, effective training programmes, user-friendly system design, and positive past experiences with technology are commonly cited as enablers. Studies also emphasise the importance of financial investment, robust IT infrastructure, and interdisciplinary collaboration in overcoming adoption challenges. For telemedicine and virtual emergency departments (VEDs), ease of use, improved accessibility, and better patient follow-up contribute to their success, whereas for AI-driven tools and clinical decision support systems (CDSS), integration with existing workflows and transparency in decision-making are crucial factors. The findings suggest that a combination of strategic planning, policy adjustments, and ongoing training can help emergency departments optimise technology adoption while mitigating resistance and operational challenges.

In order to gain a better understanding of the problem of technology adoption faced by emergency departments, this study evaluated a significant number of papers on a variety of topics relevant to emergency departments, such as telemedicine, AI-based decision support systems, virtual urgent care, and quality improvement programmes, as shown in Table 2. These evaluated papers were gathered from several countries, removing bias and demonstrating a desire to improve emergency care on a worldwide scale, including Ghana, Australia, the United States, Canada, Japan, and the United States, among other countries.

A successful adoption of technology in emergency departments requires feedback not just from doctors, but also from other emergency department stakeholders. As a result, participants in the examined studies came from a variety of backgrounds, including patients, managers, nurses, and doctors. This wide inclusion aids in providing a comprehensive picture of the problems encountered in emergency rooms as well as perceptions and experiences when it comes to various solutions. Given the importance of the data from both methodologies, this study focused on both the qualitative and quantitative research methods to ensure a complete understanding of the subject. This study provides comprehensive and patient-centred knowledge concerning how to enhance emergency medical treatment by combining a range of topics, viewpoints, countries, and research methodologies.

### 4.2. Comparative Analysis

In order to provide a more intuitive understanding of the findings, this study clusters the reviewed studies by rural and urban settings and highlights notable trends in technology adoption in emergency departments. Studies carried out in rural areas often indicate that the resource and infrastructural constraints are the main barriers to the adoption of new technologies. For example, Zachrison et al. (2020) [24] discovered that staffing shortages and costly costs are frequent barriers that rural emergency departments in the US have when attempting to utilise the telemedicine. This viewpoint was further supported by Nataliansyah et al. (2022) [30], who determined that these two obstacles were the key barriers affecting the effective use of telemedicine in emergency departments in remote areas. Furthermore, the primary barriers to the continuous application of telemedicine technologies are those related to infrastructure, such as unstable internet and telecommunications networks.

On the other hand, studies conducted in urban emergency departments, particularly those with abundant resources, typically concentrate on advanced technology like clinical decision support systems (CDSSs) and AI-based systems. For example, Fujimori et al. (2022) [22] investigated the adoption of an AI-based CDSS in a tertiary hospital. It was discovered that the key barriers faced by emergency departments in urban environments were mostly linked to usability, including incompatibility with the institution’s existing systems, which varied from those in rural areas. These urban settings benefit from more robust IT infrastructure, enabling more reliable use of these advanced technologies. Similar to this, Pu et al. (2024) [26] also conducted a study and emphasised how the metropolitan virtual emergency departments (VEDs) of Victoria, Australia, have effectively expanded because of a stronger healthcare infrastructure and a higher wealth of resources.

In summary, studies on technology adoption in emergency departments of rural hospitals have found that resource limitations, insufficient broadband availability, and staff shortages are significant barriers. On the other hand, emergency departments in urban areas with superior infrastructure and equipment are primarily related to system compatibility and user resistance rather than access to basic technology. This is due to the fact that technology is developing more rapidly in urban areas than in rural areas. Therefore, the main barriers in urban areas are mainly related to staff acceptance and the adoption of technology. However, our study has also found that financing and good management support were important facilitators in both cases. Funding could have a more significant impact in rural areas because it helps fill infrastructure gaps. Conversely, management support has a greater impact in metropolitan areas because it facilitates rapid technology adoption among employees.

Additionally, this study also grouped the reviewed studies according to the kind of technology and discovered various trends in the uptake of various technological advancements in emergency departments. For instance, AI-based systems typically face greater resistance because of their complexity. Studies conducted by Piliuk and Tomforde (2023) [47] and Fujimori et al. (2022) [22] have demonstrated that AI-driven CDSSs frequently encounter the usability issues, such as system integration problems and uncertainties regarding the precision and dependability of AI suggestions. These barriers exacerbate clinician resistance, as there is a perception that AI systems will complicate workflows and require extensive training to fully function. The complexity of these systems, coupled with concerns about system failures at critical moments in patient care, makes AI-based technologies more difficult to adopt, especially in high-pressure emergency department environments where quick decision-making is required.

In contrast, there is typically less opposition to technology like virtual emergency care systems and telemedicine platforms. Since they can expand on current communication channels and instantly increase access to care, particularly in rural or underserved regions, these systems are comparatively easy to adopt (Zachrison et al., 2020) [24]. Telemedicine solutions are thought to be useful, easy to use, and capable of rapidly improving patient outcomes. However, they still encounter obstacles, such as those related to infrastructure and worries about the calibre of virtual healthcare, but they are less formidable than sophisticated AI systems.

In conclusion, categorising emergency department technology adoption studies by rural and urban environments, as well as technology type, reveals significant tendencies across contexts. In rural emergency departments, lack of infrastructure and resources are the main barriers to technology adoption, especially telemedicine technologies, which are limited by cost, staff shortages, and unreliable communication networks. In urban settings, research focuses more on advanced technologies such as artificial intelligence (AI)-driven CDSS, but their adoption is often hindered by system compatibility and complexity in clinical use. Meanwhile, barriers to technology adoption might vary depending on the type of technology. Telemedicine and virtual emergency care systems are relatively simple to implement, as they are effective in improving access to care, especially in rural areas. In contrast, AI-based systems face more usability and integration issues, increasing resistance from clinicians.

### 4.3. Critical Analysis

This study examined 30 prior studies on technology adoption, as indicated in Table 2, in order to evaluate the challenges and enablers related to implementing new technologies in emergency rooms. By conducting this study, a detailed grasp of the elements that impact technological adoption was provided; the study also identified important facilitators that may improve adoption in these situations. Furthermore, as Table 1 shows, several studies incorporated perspectives from people who were not in emergency rooms, such as patients and hospital administrators, because these people’s opinions were thought to have an impact on how new technology is adopted in these environments.

Table 2 shows that there are several barriers that make it difficult to integrate new technology and practises in emergency departments. After analysing these studies, several common barriers that affected emergency departments’ use of technology have been found. Two of the most common barriers are high turnover rates and personnel shortages, which are frequently mentioned as major barriers. Based on the reviewed studies, it is more difficult for current teams to adapt to new technologies when there is a shortage of staff or a high employee turnover rate. This is because the instability might interrupt the continuity needed to build expertise and harness the potential of new systems. This finding is further supported by J. Y. Kim et al. (2023) [50], who conducted a study to examine the adoption of AI in healthcare. Besides that, performance and access difficulties with technology are also common barriers. Numerous studies demonstrate how personnel might become frustrated and prevent the efficient use of technology due to sluggish systems, frequent outages, and inadequate training. The same findings were proposed by Borges do Nascimento et al. (2023) [51], who carried out a study on the factors that encourage and hinder the use of digital health technologies by healthcare professionals. In addition, insufficient availability of technology, particularly in marginalised regions, may impede the optimal utilisation of novel systems. The findings of this study are further supported by J. Chen et al.’s (2023) [52] study on telehealth adoption and its limitations in rural and urban hospitals, which found that low technology availability is a key challenge, especially in rural areas. Furthermore, the combination of a high workload and resistance to change is another major barrier. Clinicians may be reluctant to accept changes when new systems or programmes are implemented because they feel overburdened by the additional responsibilities. This aversion might be exacerbated by the weight of their current responsibilities, making it more difficult to incorporate new tools into their hectic schedules. These findings align with the study conducted by Borges do Nascimento et al. (2023) [51]. They carried out a study to examine the factors that encourage and hinder healthcare workers’ use of digital health technology and found that psychological issues and heavy workloads were the main barriers.

Meanwhile, the review has identified several facilitators that can aid in the successful adoption of technology in emergency departments. One of the main facilitators is the support and involvement of management. Leaders who actively participate in the implementation process and provide the necessary resources are critical to ensuring a smooth transition. With support and commitment to the project, team confidence and morale are likely to soar, which will boost the adoption of new technology. Hall et al. (2022) [21] conducted a study that explored the designs, facilitators, barriers, and lessons learned from implementing virtual urgent care programmes, which further reinforce these findings. In addition to good leadership, cooperation and efficient communication are critical facilitators. Effective collaboration and honest communication among team members ensures that everyone is working towards the same goals and is on the same page. This collaborative approach fosters a problem-solving culture that helps resolve any difficulties that may arise during implementation. These results align with the conclusions drawn from Coffetti et al.’s (2022) [53] systematic assessment of the individual and team characteristics impacting nurses’ use of digital communication tools. Furthermore, education and training are also important enabling factors. By giving employees thorough training, employers may lessen their opposition to new technology by increasing their competence and self-assurance in utilising it. Continuous education also ensures that employees stay up to date with the latest developments, enabling them to make the most of technology. Together, these facilitators create a supportive environment that enhances the integration of new technologies in emergency departments. These findings align with a study performed by Borges do Nascimento et al. (2023) [51], which investigated the barriers and facilitators of digital health technology adoption among healthcare professionals and identified training as one of the facilitators.

From a theoretical standpoint, this study examined the facilitators and barriers associated with the adoption of technology in emergency departments in various contexts and geographies. It clarifies the barriers that influence technology adoption, such as restricted access to technology, a lack of employees, and training hurdles. Meanwhile, it has also emphasised important facilitators, such as helpful management, clear communication, and ample resources. Furthermore, the results of this study indicate that the strategy for promoting the adoption of technology should be tailored to the specific requirements of emergency departments in rural and urban areas due to the fact that various barriers are faced in different areas. For instance, expanding internet access and offering financial incentives could be more effective in rural areas. On the other hand, usability improvement and clinician engagement tactics would be more effective in urban areas. These studies have highlighted the necessity of context-specific approaches and collaborative initiatives in enhancing the effectiveness and sustainability of innovations within emergency departments. By conducting this in-depth research, this study provides healthcare professionals and policymakers with important information that can enable them to make informed decisions. This understanding can lead to targeted interventions that improve services in emergency departments, ultimately benefiting both healthcare providers and the patients they serve.

From a practical standpoint, this study suggested many tactics to improve the efficacy of technology adoption. Firstly, it is important to investigate comprehensive training programmes for employees since they may provide those employees with the confidence and information they need to effectively embrace new technology. Furthermore, the establishment of resilient support systems may efficiently tackle technical issues and offer ongoing assistance, hence reducing discontent and opposition. Thirdly, identifying and empowering local champions, which are people who support and campaign for these changes, can also greatly accelerate acceptance and guarantee that advancements are maintained over time. Healthcare organisations may improve their performance by focusing on practical strategies to overcome hurdles while deploying new technology in emergency departments.

### 4.4. Strategies for Technology Adoption

In order to better adapt to new technologies, healthcare providers should consider developing a well-established change management approach. The two most popular change management frameworks that may be employed in this circumstance are Lewin’s Change Management Model and Kotter’s eight phases for leading change [54]. These frameworks provide an organised method for handling the difficulties associated with organisational transformation. For instance, Kotter’s strategy focuses on creating a feeling of urgency among the change’s stakeholders in order to persuade them to embrace new technological developments [55]. It also brings attention to the significance of building a coalition of leaders who can effectively advocate for change, articulate their vision, and share it with the entire organisation [55]. This cooperative approach is essential for managing resistance and ensuring consensus on shared goals.

In addition, the adoption of new technologies can be accelerated through the application of Lewin’s Change Management Model. This is due to the fact that it explains the mental and emotional changes that individuals undergo throughout periods of change [56]. It defines three main processes—unfreeze, change, and refreeze—that assist firms in preparing their employees for change, making essential changes, and incorporating new practises into company culture [56]. Using these frameworks, healthcare organisations may introduce technology more methodically, addressing not just the technical issues of integration, but also the psychological and emotional factors that drive staff acceptance and engagement.

By applying a change management model and focusing on these practical strategies, healthcare organisations can enhance their operations and successfully handle the obstacles that come with integrating new technologies in emergency departments. The use of a structured change management strategy may facilitate the effective and sustainable adoption of technology by ensuring that all stakeholders are informed, involved, and provided with full assistance during the transition process.

### 4.5. Innovation and Contribution

This review study examined several studies on the adoption of technology in the healthcare industry alongside the barriers and facilitators that they discovered. The most prevalent barriers and facilitators have been identified in this study. Based on the study’s findings, the healthcare industry and technology providers may either enhance their services and products or offer solutions that address the identified facilitators and barriers. It is anticipated that this will speed up and facilitate the uptake of technology.

Emergency departments in the healthcare industry may overcome these barriers by learning from successful models in other industries. For example, several common barriers highlighted in this study may be efficiently addressed by companies that have successfully adopted and implemented technology, such as the automobile, retail and e-commerce, food, technology, and aircraft industries.

First, emergency departments may consult the automobile industry in order to address the issue of excessive turnover and a lack of personnel. By using the Standard Operating Procedure (SOP), the majority of the automotive industry has standardised procedures and streamlined workflows to minimise dependency on large teams and boost productivity. They have also introduced cross-training to provide employees with the flexibility to transition between roles [57,58]. This method can help emergency departments streamline clinical operations, cut down on unnecessary labour, and free up medical staff time for more important duties. In the face of regular staff turnover, cross-training personnel for different positions helps guarantee work continuity and efficiency.

The second issue facing emergency departments is the impact of technology on performance and accessibility. Similar challenges have also been experienced by certain retailers and e-commerce companies, such as Amazon; however, they have been able to resolve scalability and performance concerns by utilising Amazon Web Services (AWS) and other cloud computing services [59,60]. Emergency departments can similarly benefit from cloud technology to enhance system performance by minimising the system downtime, increasing data access speeds, and providing scalable storage solutions. By guaranteeing the continuous availability and stability of vital medical systems, this eventually increases user confidence in technology and pushes for its more widespread adoption.

Insufficient availability of technology is also a common issue, particularly in places with minimal resources. Emergency departments can draw inspiration from the food industry’s strategy since it has encountered comparable challenges. Starbucks is a prominent example in the food industry that has effectively surmounted this obstacle by employing a phased implementation approach [61] over several locations, all the while maintaining their conventional procedures. In underprivileged areas or marginalised regions, emergency departments should adopt a similar strategy by progressively using digital technologies. This would preserve the functioning of current systems while enabling medical staff to become accustomed to new technologies.

Resistance to change and a heavy workload are two more significant barriers to the adoption of new technologies. The technology industry, which has successfully adopted various advancements, can offer useful lessons to emergency departments. The majority of IT companies, like IBM, have discovered that including teams in decision-making enhances their ability to adapt to change and reduces resistance to new practises [62]. A similar strategy may be used by emergency departments, which would encourage cooperation and transparent communication while implementing technology gradually. This method can ease employee worries about new technologies while also resolving workload issues, thus enhancing the likelihood of effective adoption. By providing a supportive environment during the transition phase, employees will feel empowered and engaged in the process, which will further enhance the possibility of accepting changes.

Lastly, training and ongoing education are critical to the effective adoption of new technology. The healthcare industry may learn from the aircraft industry since it is most similar to the healthcare industry out of all of them. Pilots receive ongoing simulation training to assist them in becoming more competent and confident while interacting with new systems [63]. Therefore, emergency departments may guarantee that staff members are competent in using new medical equipment under pressure by providing frequent refresher courses and using simulation-based training to acquaint staff members with new technology in a low-risk setting.

### 4.6. Limitations and Future Research

The study has significant limitations. First, the number of studies analysed in this study is insufficient for providing more comprehensive and accurate findings. Larger research pools are known to provide more accurate results and more in-depth analysis. Therefore, the volume of reviewed studies plays a key role in enhancing the accuracy of research results. Conversely, a lower sample size might have an influence on the results and make it harder for them to accurately capture the distinctive features of emergency departments in the healthcare industry.

Furthermore, this study failed to make distinctions across healthcare settings, as some of the examined studies were conducted in rural locations, while others were conducted in metropolitan areas. Because of these distinctions between rural and urban areas, there may be variances in the barriers and facilitators. For example, rural areas may place a greater emphasis on the issue of the resources because they have fewer resources available, which has a significant impact on the adoption of new technologies. Since medical employees are the group of individuals who will be utilising the technology, their reluctance to change might pose a substantial obstacle to the adoption of the technology. Nevertheless, the acceptance of medical employees will be the focus in urban settings. This constraint may make it challenging to combine the results of the study or use them broadly.

Thirdly, the majority of the responses collected in these reviewed studies were self-reported by medical employees, which raises the possibility of recollection or response bias. Individuals from diverse backgrounds, such as varying roles in emergency rooms or hospitals, may contribute distinct reactions due to their exposure to disparate settings. This is because responses are influenced by institutional pressures or personal experiences, which may result in varied responses regarding barriers and facilitators.

Fourth, the lack of longitudinal data is another important limitation in this study. This is due to the fact that one-time data collection was used in the majority of the analysed studies in this study, which increases the possibility of bias. The reason for this is that it is challenging to fully understand how these technologies function over time without tracking the long-term adoption outcomes. The lack of prolonged observations may make it more challenging to precisely identify both the barriers and the facilitators of adoption since some of the important factors that influence the adoption may only become evident after and may change with continued usage. Because of this, our findings are unable to provide a comprehensive picture of the long-term dynamics related to technology adoption.

As a result, future research is encouraged to examine a wider range of studies in order to improve the accuracy and reliability of our results. Besides this, it is also important to differentiate between healthcare environments in rural and urban areas due to the notable variations in different areas in barriers and facilitators. In addition, the self-reported data should not be the only source of information used by researchers, as respondents to surveys or interviews may have inaccurate impressions of technology. Furthermore, adding longitudinal data will result in a more in-depth understanding of the long-term effects of technology adoption and a more thorough grasp of the contributing components. Thus, future studies should focus on gathering primary data from medical officers and patients in the emergency department by using different methods, including direct observation, interviews, and questionnaires. By using these methods, researchers will be able to gain firsthand data concerning technology adoption, which will help them better understand its complexities and minute changes in practical contexts. Furthermore, researching the long-term effects of technology adoption in emergency departments not only exposes changes over time but also aids in assessing its initial impact on efficiency and patient care. Longitudinal studies can identify patterns, areas of adaptation, and areas where technology use can be improved, and also will be able to provide important insights into a more complete understanding of the ultimate successful integration of technology into emergency medical settings. By addressing these questions, future research can build on existing findings and make practical recommendations to promote the widespread use of technology in healthcare.

### 4.7. Conclusions

The complicated setting of technology adoption in emergency departments is clarified in this study, which also highlights the barriers and facilitators that influence the adoption of new technologies. According to this study, there are several significant barriers that influence the adoption and leverage of new systems, such as the high turnover rates and a lack of qualified workers. This is because the instability of employees might interrupt the continuity needed to build expertise and harness the potential of new systems. Besides this, performance problems with technology, such as sluggish systems and insufficient training, exacerbate dissatisfaction and impede efficient utilisation. Furthermore, it is difficult to implement new technologies because of the excessive workload and aversion to change among practitioners, since the additional obligations might feel burdensome.

Conversely, this study identifies a few potential facilitators that may help technology adoption go more smoothly. Firstly, supportive management is essential as it can greatly increase the likelihood of effective implementation when leaders take an active role and supply the required resources. Besides this, clear communication and efficient cooperation are also essential for ensuring that everyone is in agreement and equipped to handle any problems that may come up. Last but not least, employees must receive thorough training and instruction in order to overcome their opposition to and feel comfortable when using new technologies.

In light of these findings, policymakers and healthcare leaders must develop targeted strategies using change management frameworks to address the specific challenges faced by emergency departments. In addition, policymakers should consider developing special directives that encourage collaborative efforts and tailored solutions to the unique needs of emergency departments. With these insights, the healthcare industry will be better able to encourage technological innovation, resulting in faster adoption of critical technologies and, ultimately, improved patient outcomes. On the other hand, healthcare leaders must create a culture of continuous learning and mentorship, ensuring that newer medical officers have the support they need to adapt to new technologies. Continuous learning and mentoring programmes are programmes where experienced medical officers can mentor new medical officers, reduce turnover, and increase technology utilisation. Leaders should also focus on fostering open communication and removing any organisational barriers that slow down technology adoption.

In summary, this study identified several barriers that hinder technological innovation in emergency departments. These barriers include high staff turnover, accessibility problems, a lack of available technology, opposition to change, and an excessive workload. Besides these, the study also revealed several facilitators that are capable of supporting this process, namely, collaboration, management support, and training. Based on these results, this study illustrates a few strategies to improve the adoption of technology in emergency departments, such as developing focused policies using change management frameworks, establishing a mentorship and learning culture to assist new healthcare professionals, implementing organised training programmes, and promoting open communication. However, further empirical study is needed to evaluate how these findings may be translated into practical implementation strategies, even if these results improve the understanding of technology uptake in emergency departments. Therefore, future research should further delve into these findings and investigate innovative techniques to improve technology integration and establish higher standards of care within emergency departments.

## Figures and Tables

**Figure 1 ijerph-22-00479-f001:**
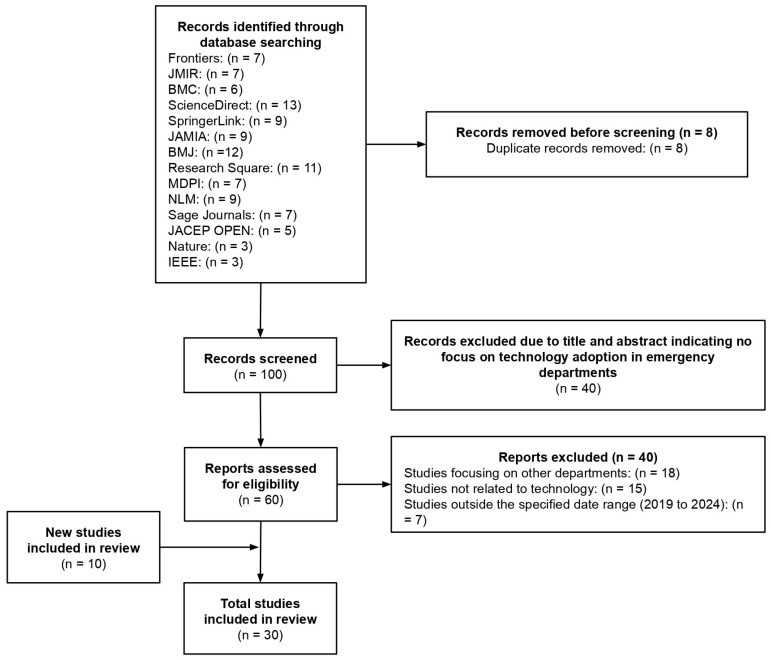
PRISMA flow diagram.

**Table 1 ijerph-22-00479-t001:** Reviewed studies.

Author	Title	Participants	Sample Size
Hall et al. (2022) [21]	Designs, facilitators, barriers, and lessons learned during the implementation of emergency department led virtual urgent care programs in Ontario, Canada	A total of 13 emergency medicine physicians and researchers with experience leading and implementing local VUC programmes	A total of 7 out of 14 VUC pilot programmes across Ontario
Fujimori et al. (2022) [22]	Acceptance, Barriers, and Facilitators to Implementing Artificial Intelligence–Based Decision Support Systems in Emergency Departments: Quantitative and Qualitative Evaluation	A total of 14 physicians from two community tertiary care hospitals in Japan Transitional year residents: 6 (0–2 years of clinical experience) Emergency medicine residents: 5 (3–5 years of clinical experience) Emergency physicians: 3 (5+ years of clinical experience)	Data from 27,550 emergency department patients from a tertiary care hospital in Japan
Anaraki et al. (2024) [14]	A qualitative study of the barriers and facilitators impacting the implementation of a quality improvement program for emergency departments: SurgeCon	Physicians, nurses, managers, patient care facilitators, programme coordinators, and patients	A total of 31 clinicians and 341 patients were surveyed via telephone
Huilgol et al. (2024) [23]	Innovation adoption, use and implementation in emergency departments during the COVID-19 pandemic	Healthcare professionals from 8 hospital-based emergency departments in the United States	49 healthcare professionals 17 doctors 7 advanced practice providers 18 nurses 7 respiratory therapists
Zachrison et al. (2020) [24]	Understanding Barriers to Telemedicine Implementation in Rural Emergency Departments	Rural Emergency Departments (EDs) in the United States	2016 NEDI-USA Survey: 977 rural EDs responded Follow-up Survey: ○ 374 rural EDs that did not use telemedicine ○ 153 rural EDs that used telemedicine
Boyle et al. (2023) [25]	Hospital-Level Implementation Barriers, Facilitators, and Willingness to Use a New Regional Disaster Teleconsultation System: Cross-Sectional Survey Study	Emergency managers from hospital-based and freestanding emergency departments (EDs) in New England states	189 hospitals and EDs were identified, with 164 (87%) responding to the survey
Pu et al. (2024) [26]	Virtual emergency care in Victoria: Stakeholder perspectives of strengths, weaknesses, and barriers and facilitators of service scale-up	Emergency medicine physicians, healthcare consumers, and other health care professionals, including residential aged care facility staff members and general practitioners	20 participants: Emergency Medicine Physicians: 10 participants Healthcare Consumers: 4 participants Other Healthcare Professionals: 6 participants (including 1 residential aged care facility staff member and 5 general practitioners)
Antor et al. (2024) [27]	Usability evaluation of electronic health records at the trauma and emergency directorates at the Komfo Anokye teaching hospital in the Ashanti region of Ghana	Trauma and emergency department staff members at Komfo Anokye Teaching Hospital	A total of 234 trauma and emergency department staff members at Komfo Anokye Teaching Hospital
Bhosekar et al. (2023) [28]	An Exploratory Study to Evaluate the Technological Barriers and Facilitators for Pediatric Behavioral Healthcare in Emergency Departments	Assistant nurse manager, nurses in charge, security accounts manager, and patient safety specialist	A total of 4 healthcare providers across two hospitals
Hodwitz et al. (2024) [29]	Healthcare workers’ perspectives on a prescription phone program to meet the health equity needs of patients in the emergency department: a qualitative study	Healthcare workers	12 interviews
Nataliansyah et al. (2022) [30]	Managing innovation: a qualitative study on the implementation of telehealth services in rural emergency departments	In total, 18 key informants from six U.S. healthcare systems (hub sites)	A total of 65 rural emergency departments (spoke sites) across 11 U.S. states
Kennedy et al. (2024) [31]	Establishing enablers and barriers to implementing the HIRAID^®^ emergency nursing framework in rural emergency departments	Emergency nurses from 11 rural, regional emergency departments in Southern New South Wales, Australia	102 nurses completed the survey
Moy et al. (2023) [32]	Understanding the perceived role of electronic health records and workflow fragmentation on clinician documentation burden in emergency departments	Physicians and registered nurses	24 responses: 12 physicians 12 registered nurses
Uscher-Pines et al. (2021) [33]	Rising to the challenges of the pandemic: Telehealth innovations in U.S. emergency departments	A total of 15 emergency department leaders from 14 institutions across 10 states in the United States	A total of 35 individuals were invited to participate, resulting in a response rate of 43%
Wong et al. (2024) [34]	Formative evaluation of an emergency department clinical decision support system for agitation symptoms: a study protocol	Emergency department physicians, nurses, technicians, and patients with lived experience of restraint use during an emergency department visit	Sample Size for Initial Design: Approximately 5–6 focus groups with 6 participants each Usability Testing: Around 15 participants in total, across three rounds of refinement Field Testing: Observation of eight patient encounters Pilot Trial: At least 26 eligible subjects
Barton et al. (2024) [35]	Academic Detailing as a Health Information Technology Implementation Method: Supporting the Design and Implementation of an Emergency Department–Based Clinical Decision Support Tool to Prevent Future Falls	Emergency medicine resident physicians and advanced practice providers	16 participants (10 resident physicians and 6 advanced practice providers)
Billah et al. (2022) [36]	Clinicians’ perspectives on the implementation of patient decision aids in the emergency department: A qualitative interview study	Emergency clinicians, including attending physicians, resident physicians, and physician assistants	20 emergency clinicians
Davison et al. (2024) [37]	Barriers and facilitators to implementing psychosocial digital health interventions for older adults presenting to emergency departments: a scoping review protocol	The scoping review considers articles that include older adults (70 years and older) who received care in an emergency department setting, as well as other stakeholders such as patient families, clinical staff, and other hospital staff involved in the care of older adults in EDs	The review will include both qualitative and quantitative studies, but the exact sample size will depend on the studies identified and included in the review
Salwei et al. (2022) [38]	Usability barriers and facilitators of a human factors engineering-based clinical decision support technology for diagnosing pulmonary embolism	Emergency medicine physicians	32 emergency medicine physicians: 8 year 1 residents 8 year 2 residents 8 year 3 residents 8 attending physicians
Simpson et al. (2023) [39]	Implementation strategies to address the determinants of adoption, implementation, and maintenance of a clinical decision support tool for emergency department buprenorphine initiation: a qualitative study	Clinicians from five different healthcare systems, including the following: Attending doctors Physician assistants Residents	28 interviews
Shin et al. (2024) [40]	Barriers and Facilitators to Using an App-Based Tool for Suicide Safety Planning in a Psychiatric Emergency Department: A Qualitative Descriptive Study Using the Theoretical Domains Framework and COM-B Model	Nurses, psychiatrists, social workers, programme assistants, and chemists	29 emergency department professionals
Shuldiner et al. (2023) [41]	The Implementation of a Virtual Emergency Department: Multimethods Study Guided by the RE-AIM (Reach, Effectiveness, Adoption, Implementation, and Maintenance) Framework	Patients utilizing the virtual emergency department (VED) and medical specialists	Average of 153 visits per month
Sharifi Kia et al. (2023) [42]	Telemedicine in the emergency department: an overview of systematic reviews	Review-based study	Analysis of 18 studies (not direct participant data)
Hose et al. (2023) [43]	Work system barriers and facilitators of a team health information technology	Professionals from 12 different healthcare disciplines	36 healthcare workers
Tyler et al. (2024) [44]	Use of Artificial Intelligence in Triage in Hospital Emergency Departments: A Scoping Review	Review-based study	29 publications selected from an initial 1142 reviewed
Hudson et al. (2023) [45]	Perspectives of Healthcare Providers to Inform the Design of an AI-Enhanced Social Robot in the Pediatric Emergency Department	Medical professionals from 2 paediatric emergency departments in Canada	11 medical professionals
Katzman et al. (2023) [46]	Artificial intelligence in emergency radiology: A review of applications and possibilities	Review-based study	44 studies reviewed
Piliuk and Tomforde (2023) [47]	Artificial intelligence in emergency medicine. A systematic literature review	Review-based study	116 studies reviewed
Jordan et al. (2023) [48]	The Impact of Cultural Embeddedness on the Implementation of an Artificial Intelligence Program at Triage: A Qualitative Study	Triage nurses in a community hospital’s emergency department in the United States	13 triage nurses
Talevski et al. (2024) [49]	From concept to reality: A comprehensive exploration into the development and evolution of a virtual emergency department	Patients using the Victorian Virtual Emergency Department in Victoria, Australia	500 patients who used the VVED service

**Table 2 ijerph-22-00479-t002:** Summary of studies.

Author	Title	Barriers	Facilitators
Hall et al. (2022) [21]	Designs, facilitators, barriers, and lessons learned during the implementation of emergency department led virtual urgent care programs in Ontario, Canada	Behaviour change Technology access Equitable access standardised data collection	Local champions Provincial funding Patient engagement Effective marketing strategies
Fujimori et al. (2022) [22]	Acceptance, Barriers, and Facilitators to Implementing Artificial Intelligence–Based Decision Support Systems in Emergency Departments: Quantitative and Qualitative Evaluation	System performance Compatibility	Evidence strength Design quality
Anaraki et al. (2024) [14]	A qualitative study of the barriers and facilitators impacting the implementation of a quality improvement program for emergency departments: SurgeCon	Low manager participation and autocratic leadership Staff shortages and high turnover rates Poor communication and lack of teamwork Negative perceptions from past failures Resistance to change and heavy workloads	Supportive and engaged management Sufficient staffing and resources Effective communication within and between departments Positive experiences with past interventions High motivation for improvement
Huilgol et al. (2024) [23]	Innovation adoption, use and implementation in emergency departments during the COVID-19 pandemic	Lack of evidence-based information Evolving guidelines Anxiety Moral distress Clinician resistance	Social media Clinician autonomy Organisational culture Supportive leadership External experiences
Zachrison et al. (2020) [24]	Understanding Barriers to Telemedicine Implementation in Rural Emergency Departments	Cost Staffing issues Transfer patterns Perceived complexity of telemedicine systems	Support from emergency department leadership Support from hospital or health system leadership
Boyle et al. (2023) [25]	Hospital-Level Implementation Barriers, Facilitators, and Willingness to Use a New Regional Disaster Teleconsultation System: Cross-Sectional Survey Study	Limited access to specialists Credentialing delays Unreliable internet and cellular service	Adequate emergency notification systems Strong telecommunication infrastructure High willingness to adopt
Pu et al. (2024) [26]	Virtual emergency care in Victoria: Stakeholder perspectives of strengths, weaknesses, and barriers and facilitators of service scale-up	Difficulty in conducting thorough physical assessments virtually Lack of robust evidence and standardised guidelines for Virtual Emergency Departments (VEDs) Additional workload and resource constraints faced by emergency department staff	Convenience and accessibility Improved coordination and follow-up care Enhanced safety
Antor et al. (2024) [27]	Usability evaluation of electronic health records at the trauma and emergency directorates at the Komfo Anokye teaching hospital in the Ashanti region of Ghana	Insufficient system training and malfunctions Power outages Privacy concerns Insufficient maintenance	Comfort in using the system System reliability Improved patient care quality
Bhosekar et al. (2023) [28]	An Exploratory Study to Evaluate the Technological Barriers and Facilitators for Pediatric Behavioral Healthcare in Emergency Departments	Software usability issues Inadequate training Quality concerns	Accessibility of clinician notes Effective safety tools Efficient communication systems
Hodwitz et al. (2024) [29]	Healthcare workers’ perspectives on a prescription phone program to meet the health equity needs of patients in the emergency department: a qualitative study	Unclear eligibility criteria Inconsistent phone availability Potential bias in administration	Non-judgemental, anti-oppressive approach by healthcare staff Smooth enrolment process Deep understanding of marginalised patients’ needs
Nataliansyah et al. (2022) [30]	Managing innovation: a qualitative study on the implementation of telehealth services in rural emergency departments	Lack of adequate needs assessments Insufficient training for local staff, leading to unsuccessful consultations Resource limitations in providing consistent telehealth services Political and regulatory challenges	Comprehensive needs assessments to tailor services to regional needs Thorough staff training programmes for effective technology use Strong connections between hub and spoke locations for cooperation and trust Sufficient service capacity and backup mechanisms for reliable delivery
Kennedy et al. (2024) [31]	Establishing enablers and barriers to implementing the HIRAID^®^ emergency nursing framework in rural emergency departments	Time and resource constraints Lack of awareness and understanding of the framework Limited management support Uncertainty about the framework’s efficacy	Nurses’ enthusiasm for new approaches Belief in HIRAID^®^’s potential to improve nursing practises and patient outcome
Moy et al. (2023) [32]	Understanding the perceived role of electronic health records and workflow fragmentation on clinician documentation burden in emergency departments	Inadequate functionality of EHRs Poorly designed user interface causing increased manual labour Workflow disruptions due to frequent task-switching and interruptions Increased cognitive strain and reduced documentation quality	Incorporating patient-specific EHR features Minimising manual data entry using advanced data collection techniques Improving EHR displays and setups for better usability
Uscher-Pines et al. (2021) [33]	Rising to the challenges of the pandemic: Telehealth innovations in U.S. emergency departments	Need for direct support within the emergency department Technological challenges	Higher compensation for staff Relaxed licensure requirements Loosened HIPAA regulations
Wong et al. (2024) [34]	Formative evaluation of an emergency department clinical decision support system for agitation symptoms: a study protocol	Time constraints in managing agitation Inconsistent workloads Limited access to professional psychiatric evaluations	User-centred design with continuous end-user feedback Proactive interventions Systematic risk assessments
Barton et al. (2024) [35]	Academic Detailing as a Health Information Technology Implementation Method: Supporting the Design and Implementation of an Emergency Department–Based Clinical Decision Support Tool to Prevent Future Falls	Hectic atmosphere of the emergency room Physicians’ misinterpretations of the CDS tool or the referral procedure Lack of clarity in the referral process	Ease of use of the tool Minimal input requirements Automated identification of high-risk patients Academic detailing interviews to clarify misconceptions and enhance understanding
Billah et al. (2022) [36]	Clinicians’ perspectives on the implementation of patient decision aids in the emergency department: A qualitative interview study	Patients’ worries, such as inadequate health literacy Uncertainties about the validity of the Das Concerns about increased medicolegal risk Perception that DAs are unnecessary Lack of time to apply DAs	Positive attitudes towards shared decision-making (SDM) Patient access to follow-up care Possibility of increased patient satisfaction Better risk communication Efficient integration of DAs into clinical workflows Institutional support
Davison et al. (2024) [37]	Barriers and facilitators to implementing psychosocial digital health interventions for older adults presenting to emergency departments: a scoping review protocol	Complexity of health problems older adults face Urgent need for care Possibility that older adults are not tech-savvy	Acceptance of older individuals using portable electronic devices (PEDs) Potential for digital psychological therapies to improve care and reduce unforeseen hospital visits
Salwei et al. (2022) [38]	Usability barriers and facilitators of a human factors engineering-based clinical decision support technology for diagnosing pulmonary embolism	Incompatibility with certain workflows Preference for single risk assessments or simultaneous orders Limited support for collaboration between residents and attending physicians	Automatic display of vital signs Time-saving in finding information Support in ordering diagnostic tests and generating paperwork
Simpson et al. (2023) [39]	Implementation strategies to address the determinants of adoption, implementation, and maintenance of a clinical decision support tool for emergency department buprenorphine initiation: a qualitative study	Clinical training Organisational culture Patient referrals for further care Customisation of implementation for each emergency department	Efforts of local activists Effective training programmes Supportive environment Efficient referral tools Comprehensive, multilevel approach for implementation
Shin et al. (2024) [40]	Barriers and Facilitators to Using an App-Based Tool for Suicide Safety Planning in a Psychiatric Emergency Department: A Qualitative Descriptive Study Using the Theoretical Domains Framework and COM-B Model	Issues with paperwork and communication Cell phone access issue for patients Lack of connectivity between the app and existing electronic health record systems	Enhanced patient accessibility and care efficiency Strong professional identity and duty driving utilisation of the app
Shuldiner et al. (2023) [41]	The Implementation of a Virtual Emergency Department: Multimethods Study Guided by the RE-AIM (Reach, Effectiveness, Adoption, Implementation, and Maintenance) Framework	Scheduling difficulties for follow-up appointments Concerns from medical specialists about the efficiency of remote consultations	Safe and quiet environment for virtual visits Quick and convenient access to treatment High levels of patient satisfaction Iterative approach allowing adaptation to patient and provider needs
Sharifi Kia et al. (2023) [42]	Telemedicine in the emergency department: an overview of systematic reviews	Poor infrastructure and technological challenges Insufficient evidence supporting the viability of telemedicine in EDs	-
Hose et al. (2023) [43]	Work system barriers and facilitators of a team health information technology	Technical issues (system incompatibilities, software malfunctions) Resistance to change among staff and inadequate training	Strong leadership and organisational commitment to technology Adequate funding and comprehensive training programmes
Tyler et al. (2024) [44]	Use of Artificial Intelligence in Triage in Hospital Emergency Departments: A Scoping Review	Staff resistance to adopting new technologies Concerns regarding data security and privacy High initial costs associated with AI implementation Need for transparent and understandable algorithms	-
Hudson et al. (2023) [45]	Perspectives of Healthcare Providers to Inform the Design of an AI-Enhanced Social Robot in the Pediatric Emergency Department	Age-appropriate interactions Space constraints in EDs Capacity of robots for independent situational adaptation	Individualised emotional support for children Positive reinforcement by the robots
Katzman et al. (2023) [46]	Artificial intelligence in emergency radiology: A review of applications and possibilities	High costs Need for medical education Doubts over algorithm reliability	Financial investment Cultural change in medical practice
Piliuk and Tomforde (2023) [47]	Artificial intelligence in emergency medicine. A systematic literature review	High cost of AI Extensive training requirements Data privacy concerns Unequal AI performance in different settings	-
Jordan et al. (2023) [48]	The Impact of Cultural Embeddedness on the Implementation of an Artificial Intelligence Program at Triage: A Qualitative Study	Aversion to change Reliance on clinical expertise Scepticism about AI’s ability to consider contextual and cultural elements	Enhanced patient safety Efficient decision-making support Thorough training and easy integration into workflows
Talevski et al. (2024) [49]	From concept to reality: A comprehensive exploration into the development and evolution of a virtual emergency department	Technical difficulties Resistance from healthcare professionals Concerns over the quality of virtual care	Strong institutional backing Advances in digital healthcare Increased demand due to COVID-19

## Data Availability

This study is not affected by ethical issues surrounding data security, privacy, or confidentiality because of the nature of the review and the lack of human participants in the research process.

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
