# Peer review of "Barriers to and Facilitators of Technology Adoption in Emergency Departments: A Comprehensive Review"

_ijerph, 2025, doi:10.3390/ijerph22040479_

Round 1
Reviewer 1 Report
Comments and Suggestions for Authors
Dears Authors,
I read the comprehensive study titled ''Barriers and Facilitators of Technology Adoption in Emergency 2 Departments A Comprehensive Review''. There is only unnecessary general information in the introduction part of the article. I think it could be shortened a bit. Although this article is very long and contains a lot of information, It summarizes the many studies published in the literature on this subject very well. I think it will contribute to both the readers and science
yours sincerely
Reviewer 2 Report
Comments and Suggestions for Authors
- The article is well structured
- The references are well cited
- The literatura review and material methods are pretty well presented
- Discussions strongly confirm this review
-
The subject explored is interesting and brings a new perspective in the ER, especially with the current trend of medical technologization. The main area that is analyzed by this study is the adoption of technology in the ER. The subject is interesting and quite new, proved by the wide variety of participants in each study and by the limited amount of scientific literature available, especially in the case of the Emergency Department. It seems relevant to the field, with few reviews being available. With that being said, I feel that there are some improvements to be made:
About methodology
-a better exclusion of articles might be necessary with some of them having as little as 4 participants and some having up to 27550
About improvements regarding the review itself
-the comparison between urban and rural areas seems interesting, but a more in-depth analysis of it, I think, would benefit the manuscript, this analysis might prove interesting
-the table with the studies selected should be positioned in landscape format for better visualization
-the initial part (The literature review) seems pointless, bearing in mind the fact that the same information is discussed again in Discussion and in Results as well
The conclusions seem adequate for this review, with the data that has been described, but a bit exaggerated, since it does not describe how ERs should adapt, it only describes summaries of other emergency departments. A tone down of the conclusions might be a better fit. The references seem appropriate, with every one of them being cited correctly.
Reviewer 3 Report
Comments and Suggestions for Authors
I would like to thank the authors for their effort to explore the barriers and facilitators to implementing technologies that can improve patient care in emergency services by reviewing the state of the art on the subject.
I would like to kindly make some contributions:
- Abstract: the method must include the design of your study, comprehensive review and context, criteria for the review, and the objective and hypothesis. The results are the studies you addressed and their characteristics.
- Introduction: I suggest you specify the impact of the use of health technologies in emergency services, specific examples of them and how they could improve the living conditions of health workers and/or patients, and that you clearly state your research question for this review.
- In section 2, literature review, what the authors do is an interpretation of their results; it would be convenient if you placed this section appropriately in its appropriate place.
- Methods: I am not clear about the criteria you adopted to guarantee the quality of the study. I find this section difficult to read. I think it would be easier if you started by indicating the databases in which you performed your searches, with which terms and if you took into account any inclusion and/or exclusion criteria.
- The results section is missing, in which you should also include table 1.
- Conclusions: I recommend summarizing it and giving an answer based on the research question and objective of the study, identifying those barriers and facilitators.
- Discussion: you should include references to other research similar to yours and compare it with your findings.
Round 2
Reviewer 3 Report
Comments and Suggestions for Authors
I thank the authors for taking our contributions into consideration.